# *Paenibacillus polymyxa* A26 and Its Surfactant-Deficient Mutant Degradation of Polycyclic Aromatic Hydrocarbons

**Salme Timmusk [1,\*], Tiiu Teder [2] and Lawrence Behers [3]**

[1] Department of Forest Mycology and Plant Pathology, Swedish University of Agricultural Sciences, P.O. Box 7026, SE-75007 Uppsala, Sweden
[2] Bashan Institute of Science, 1730 Post Oak Ct., Auburn, AL 36830, USA; tiiuteder@gmail.com
[3] Novawest Communications and Technologies, Tucson, AZ 85715, USA; novawest@aol.com
[\*] Correspondence: salme.timmusk@slu.se

**Abstract:** We compared the ability of two bacterial strains, *Paenibacillus polymyxa* A26 and *P. polymyxa* A26Sfp, for biodegradation of naphthalene (NAP). The studies were performed under simulated laboratory conditions, in liquid medium and soil with different carbon sources, pH and salt contents. Changes in the luminescence inhibition of *Aliivibrio fischeri*, as an indicator of the baseline toxicity, were observed in degradation mixtures during 7 days of incubation. While both strains expressed the best growth and NAP degradation ability in the minimal salt medium containing sucrose and 5% NaCl at pH 7 and 8, the mutant strain remained effective even under extreme conditions. A26Sfp was found to be an efficient and potentially industrially important polycyclic aromatic hydrocarbon degradation strain. Its extracellular polysaccharide production is 30%, and glucan production is twice that of the wild type A 26. The surface tension reduction ability was ascertained as 25–30% increased emulsification ability.

**Keywords:** polycyclic aromatic hydrocarbons; biodegradation; *Paenibacillus polymyxa* A26; Sfp-type 4-phosphopantetheinyl transferase; exopolysaccharides





## 1. Introduction

Crude oils contain many major and minor constituents. The properties of the constituents influence how the spilled oil behaves and determines the fate and effects of the spill in the environment [1]. Among the components of crude oil, polycyclic aromatic hydrocarbons (PAHs) are a class of organic contaminants that contain two or more fused benzene rings; they are considered important after oil spill accidents because they are toxic, mutagenic, carcinogenic, and relatively persistent in the environment [2,3]. Of the various forms of PAHs in crude oil, 16 PAHs including naphthalene (NAP) are considered as priority pollutants by the environmental protection agencies [4–6].

Many studies have focused on the isolation of bacteria that produce biomolecules that promote biodegradation of the PAH and other pollutants and that remove pollutants from the environment [7–10]. Our wild barley (*Hordeum spontaneum*) rhizosphere isolate *Paenibacillus polymyxa A* 26 originates from a habitat exposed to various stress factors at the Evolution Canyon (EC) South Facing Slope (SFS) [11] (Table 1 and Figure S1 [11]). The *P. polymyxa* A26 Sfp-type 4-phosphopantetheinyl transferase deletion mutant strain (A26Sfp) enhanced plant drought stress tolerance [12]. The mutant, compared to its wild type A26, is 30% enhanced in its biofilm exopolysaccharide (EPS) production [13–15]. This correlates with the improved drought stress tolerance conferred by the strain and the enhanced biocontrol ability [12–16]. *P. polymyxa* is a bacterium widely used in agriculture, industry, and environmental remediation because it has multiple functions [14,16–24]. *P. polymyxa* strains from the harsh South Facing Slope (SFS) in comparison to the moderate North Facing Slope (NFS) at 'Evolution Canyon' (EC), Israel, show huge differences in their metabolism, drought tolerance enhancement and biocontrol ability [25]. Our *P. polymyxa*

strain A26 is isolated from the stressful SFS and has been shown to be capable of moderate drought stress tolerance enhancement [25]. An important pool of the bioactive compounds of great interest for biotechnology are non-ribosomal peptides/polyketides. The spectrum of application of both classes of compounds is large. Non-ribosomal peptides produced by non-ribosomal peptide synthetases (NRPS), and polyketides are produced by polyketide synthetases (PKS). Both are diverse families of natural products with an extremely broad range of biological activities [12,26] These molecules exhibit a broad range of structural diversity and display biological activities that include adaptation to unfavourable environments, and communication or competition with other microorganisms in their natural habitat [12,26].

**Table 1.** Strains used in the study.

| Name | Origin | Publications |
|---|---|---|
| Paenibacillus polymyxa A26 | Wild barley rhizosphere, the Evolution Canyon, Haifa, Israel | [25] |
| Paenibacillus polymyxa A26Sfp | Wild barley rhizosphere, the Evolution Canyon, Haifa, Israel | [12,27] |

The *A. fischeri* bioluminescence inhibition test constitutes a simple and economic technique, frequently applied for ecotoxicological screening [28]. The test is sensitive, easy to apply and reproducible, thereby facilitating testing for the ecotoxicity screening of different compounds. Several commercially available types of this assay have been developed to determine the acute and chronic toxicities of diverse chemicals.

In the present investigation, we compared the intrinsic ability of two bacterial strains, A26 and its mutant A26Sfp, for degradation of a polycyclic aromatic hydrocarbon NAP in a defined mineral medium and axenic soil. While for the neutral and slightly alkaline environment (pH 7 and 8), sucrose and 5% NaCl in the medium were most favourable for both strains, the mutant showed 20–35% higher biodegradation efficiency and could function under extreme conditions. We explored a possible link, involving improved emulsification between the strain's biodegradation ability and the bacterial EPS production.

## 2. Material and Methods

### 2.1. Bacterial Growth, Culture Conditions and Chemicals

*P. polymyxa* A26 originates from the wild barley rhizosphere from the SFS at EC (Figure S1 [25] Table 1). The *P. polymyxa* A26 Sfp-type 4-phosphopantetheinyl transferase deletion mutant strain (A26Sfp) was generated as previously described [12,27]. Overnight cultures of the bacterial strains in TSB were washed and resuspended in modified *DF* salt minimal *medium* (MM) (A600, 0.6) [29]. The composition of DF salts minimal medium was: 4 g $KH_2PO_4$, 6 g $Na_2HPO_4$, 0.2 g $MgSO_4$. $7H_2O$, 1 mg $FeSO_4$. $7H_2O$, 10 µg $H_3BO_3$, 10 µg $MnSO_4$, 70 µg $ZnSO_4$, 50 µg $CuSO_4$, 10 µg $MoO_3$, 15 g glucose, 2 g gluconic acid, 2 g citric acid and 1000 mL distilled water. A volume of 10 mL of the resuspension was inoculated into Erlenmeyer flasks and incubated in a 200 rpm shaker for 24 h at 28 ± 2 °C. Furthermore, the NAP (100 mg/L) were added individually to the culture flasks and incubated in the dark at 200 rpm 28 ± 2 °C for predetermined time periods.

The polycyclic aromatic hydrocarbon naphthalene (NAP), kerosene, hexadecane, were purchased from Sigma Aldrich (St. Louis, MO, USA).

### 2.2. NAP Degradation in Culture Medium

The bacterial cultures were prepared as described above and culture flasks were incubated at 200 rpm for 8 days at 28 ± 2 °C. Bacterial growth was detected by measuring the optical density at 600 nm daily during 8 days of growth. All experiments were performed in triplicate. Growth in media without NAP were used as controls.

### 2.3. Analysis of NAP Degradation Efficiency

Changes in the luminescence inhibition of *Aliivibrio fischeri*, as an indicator of the baseline toxicity, were observed in photodegradation mixtures. The BioTox™ Kit was used for the determination of toxicity of samples. The inhibitory effect of the sample on the light emission of luminescent bacteria, *A. fischeri* (formerly *Vibrio fischeri*), was measured with a luminometer.

We reconstituted lyophilised aliquots of *A. fischeri* containing NaCl (3% $w/v$) by adding 1 mL of distilled water and resuspended them in 10–30 mL of the nutrient broth. Two hundred microliters of the bacteria suspension and 100 μL of each sample were added to the microplate wells. The controls consisted of 200 μL of the bacteria plus 100 μL of a 3% NaCl solution in tap water. The emitted light was recorded by a Victor Light 1420 microplate luminometer (Perkin-Elmer, Norwalk, CT, USA) at fixed intervals between 0 and 48 h. Five replicates were prepared for each sample, and the light emission values expressed as relative luminescence units (RLU). Biodegradation efficiency was calculated by measuring the luminescence of the bacteria after 30 min of contact with the contaminant. To determine the toxic effect, we compared the emitted light from the samples at the various dilutions with the control solution. The less light emitted, the greater the toxicity of the sample. Therefore, the relative biodegradation efficiency of the bacterial strain is expressed as the percentage inhibition of the sample (I% sample) divided from the control sample with the NAP where the bacteria were not used (I% control). The biodegradation efficiency percentage (BE%) was used to express the toxicity of the tested samples and calculated according to: BE% = I% control − I% sample

### 2.4. Effect of Different Carbon Sources on NAP Degradation

The cultures were prepared as described above except that five different carbon sources, including glucose, galactose, fructose, maltose, and sucrose, were added individually into MM medium at a concentration of 1.5%. Culture flasks were incubated in a 200 rpm shaker for 7 days at 28 ± 2 °C Samples were withdrawn daily and NAP degradation efficiency of the culture was determined by luminescence inhibition (using the BioTox kit as described above).

### 2.5. Effect of pH on NAP Degradation

Batch experiments were performed to study the effect of pH on NAP biodegradation.

The cultures were prepared as described above except that the pH of the culture medium was adjusted from 5 to 10 and the flasks were incubated in a 200rpm shaker for 7 days at 28 ± 2 °C. Samples were withdrawn daily and luminescence inhibition was determined.

### 2.6. Effect of NaCl on NAP Degradation

The effect of NaCl on biodegradation of NAP was performed in batch experiments. The cultures were prepared as described above except that NaCl (1–15%) was added individually to MM bottles. Cultures were incubated in a 200 rpm shaker for 7 days at 28 ± 2 °C. Samples were withdrawn daily, and NAP degradation efficiency of the culture was determined by luminescence inhibition.

### 2.7. EPS Extraction

EPS extraction was performed as described earlier with small modifications [30]. Briefly, bacterial cultures were diluted 1:5 with distilled water and centrifuged for 30 min at 17,600× $g$ at 20 °C to separate cells. Then, EPS were precipitated by slowly pouring the supernatant into two volumes of isopropanol while stirring at 200 rpm. The filtered polysaccharide was suspended in a digestion solution consisting of 0.1M $MgCl_2$, 0.1 mg/mL DNase, and 0.1 mg/mL RNase solution, and incubated for 4 h at 37 °C. Samples were extracted twice with phenol-chloroform and lyophilised using a Virtis SP Scientific 2.0

freeze dryer. For the emulsification, drop collapse, and oil spreading assay, the dialysed EPS were taken into the initial volume in double distilled sterile water.

### 2.8. Glucan Production Assay

The bacterial strains were grown on TSB, washed, and resuspended in MM medium (A600, 0.6). A volume of 10 mL of the resuspension was inoculated into Erlenmeyer flasks and incubated in a 200 rpm shaker for 5 days at 28 ± 2 °C. The glucan production assay was performed as described earlier [31]. Briefly, after incubation, glucan was recovered by adding an equal volume of 0.6N NaOH with 30 min stirring to the production medium. The bacterial cells were removed from the production medium by centrifugation at 10,000 rpm for 10 min. The supernatant was neutralized to pH 7 by adding 4N acetic acid. Then, the material was washed with water until the pH was neutralized, causing the precipitation of the curdlan, which was thereafter lyophilised.

### 2.9. Emulsification Assay

The emulsification assay used was previously described by Cooper and Goldenberg [32]. Overnight cultures of the bacterial strains in TSB were washed and resuspended in MM medium (A600, 0.6). A volume of 10 mL of the resuspension was inoculated into Erlenmeyer flasks and incubated in a 200 rpm shaker at 28 ± 2 °C for 7 days. Cell-free culture broth/bacterial EPS or glucan (200 μL) was used to determine the emulsification of NAP in an Eppendorf tube containing 600 μL of distilled water. A volume of 1.2 mL of NAP was mixed with each sample in triplicate. For two min, the mixture was vortexed, and the emulsion was allowed to stand for 24 h. Water and NAP served as negative controls. The height of the emulsion layer was then measured. The emulsification index was calculated based on the ratio of the height of the emulsion layer and the total height of the liquid [EI % = (emulsion/total h) × 100]. To determine the stability of the emulsification ability of the biosurfactant, the emulsification index was also determined after 5 and 7 days.

### 2.10. Drop-Collapse Test

Both bacterial strains' culture filtrates and A26Sfp EPS and glucan extracts were submitted to a drop-collapse test, using the procedure described by Jain et al. [33]. For a period of 7 days the strains were grown in MM medium as described above, and 5 μL of the cell-free culture broth (supernatant centrifuged 13,000× *g*, 15 min) was dropped on a glass slide covered with crude oil. The result was considered positive for biosurfactant production when the drop diameter was at least 1 mm larger than that produced by distilled water (negative control).

### 2.11. Oil Spreading Assay

A volume of 10 μL of crude oil was placed on the surface of a Petri dish that contained 40 mL of distilled water. A thin layer of oil was formed, as described earlier [9]. Culture supernatants, EPS, and glucan extracts (10 μL, obtained as above) were then placed in the centre of the oil layer. If the oil is displaced by an oil-free clearing zone, then biosurfactant is present in the supernatant. A negative control was performed with distilled water (without biosurfactant), and no oil displacement or clear zone was observed.

### 2.12. NAP Degradation in Soil

Sterile uncontaminated peat soil was artificially contaminated by adding the defined NAP mixture, prepared in DCM, to a sterile jar, allowing the solvent to evaporate, and then the soil was added to the jar. After thorough mixing, the homogeneity of NAP distribution was confirmed by testing using the luminescence inhibition assay in five random samples of the soil. The soil was subdivided into 200 g (dry weight) lots of 1.5-L jars. The bacterial cultures were prepared as described above, and Erlenmeyer flasks were incubated in a 200 rpm shaker for 24 h at 28 ± 2 °C. The jars were then inoculated to provide a bacterial population of $10^6$ cells per g of soil. Controls, which contained NAPs but lacked inoculum,

were set up similarly. All soil cultures were supplemented with sterile MM solution with 5% NaCl, 1.5% sucrose, pH 8 to approximately 65% of the soil's water-holding capacity and were incubated at 25 °C in the dark.

Three samples of 1 g of soil from each jar were collected daily during the 7 days of incubation for analysis of NAP biodegradation and measurement of the microbial population as described by us earlier [34]. Briefly, aliquots of 10 mM of primers 1492R (5-GGTTACCTTGTTACGACTT-39') and 27F (5'-AGAGTTTGATCCTGGCTCAG-3') and 1 mL of template were used. The reaction was performed in 10 mL PCR mix. The reaction conditions were 95 °C for 2 min, followed by 30 cycles of denaturation at 95 °C for 15 s, annealing at 55 °C for 20 s, primer extension at 72 °C for 1 min, followed by the final extension at 72 °C for 5 min. For sequencing, the PCR products were purified with a QIAquickTM Gel Extraction kit (QIAGEN, Hilden, Germany).

### 2.13. Data Confirmation and Validation

To ensure reproducibility, each treatment was conducted in triplicate, and the entire experiment was repeated three times. Replicated data were studied for normal distribution and analysed by MiniTab17 ANOVA. The treatment effects were considered statistically significant, $p \leq 0.01$ (indicated by different letters) or nonsignificant. Univariate analysis of selected samples involving distribution and variability of distribution was performed using the Unscrambler X10.4.1 descriptive statistics.

## 3. Results

### 3.1. A26 and A26Sfp NAP Degradation Ability Is Dependent on the Medium pH, Salinity and C Source

The growth of A26 and A26Sfp in the presence and absence of NAPs at an initial concentration of 100 mg/L was determined as optical density at 600 nm (Figure 1). The time course indicates that both strains are capable of utilizing NAP. It is often reported that biodegradation is dependent on survival, characterized by ODs. However, this measure would also indicate adaptation. Hence, biodegradation was analysed using the bioluminescence inhibition test (Figure 2). Generally, microorganisms require suitable growth conditions (e.g., carbon sources, nutrients, pH) since these strongly affect their growth.

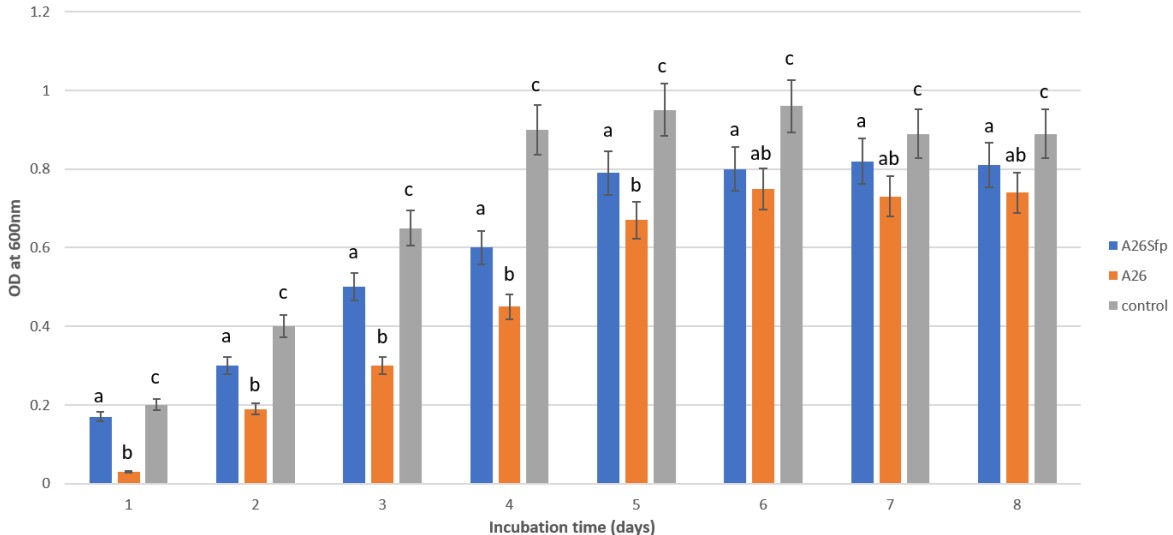

**Figure 1.** Growth (optical density at 600 nm) of *Paenibacillus polymyxa* A26 *and P. polymyxa* A26Sfp in mineral medium containing naphthalene (100 mg/L). Different letters indicate statistically significant differences; see Material and Methods.

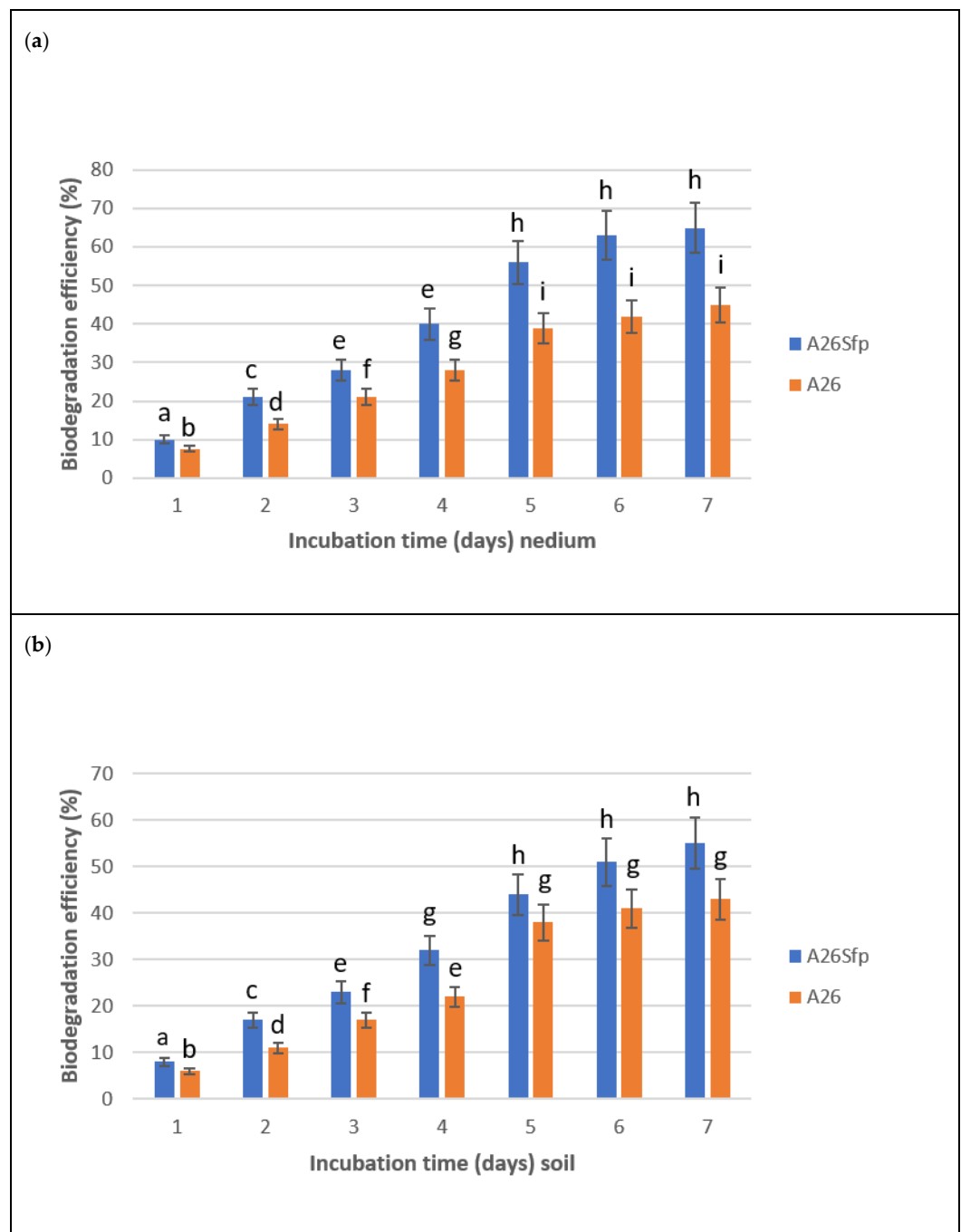

**Figure 2.** Degradation potential of *Paenibacillus polymyxa* A26 and *P. polymyxa* A26Sfp exposed to 100 mg/L of naphthalene in mineral medium containing sucrose (**a**) and in soil (**b**). Different letters indicate statistically significant differences; see Material and Methods.

Our results show that maximum degradation efficiency was reached at the 5th day. A26Sfp and A26 efficiently degraded NAP by 69% and 49%, respectively, in 5 days of incubation in liquid medium and soil (Figure 2a,b). Somewhat reduced numbers in the degradation efficiency in the case of degradation in soil (55 and 44%) were observed (Figure 2a,b). Our results show that the two strains used in the study perform best in the medium supplemented with sucrose (Figure 3) at pH 7 and 8 (Figure 4) in 5% NaCl (Figure 5). While the A26 biodegradation efficiency was significantly reduced at pH 5, 6, and at 15% NaCl, the mutant remained effective even at extreme conditions of pH and salinity (Figures 4 and 5).

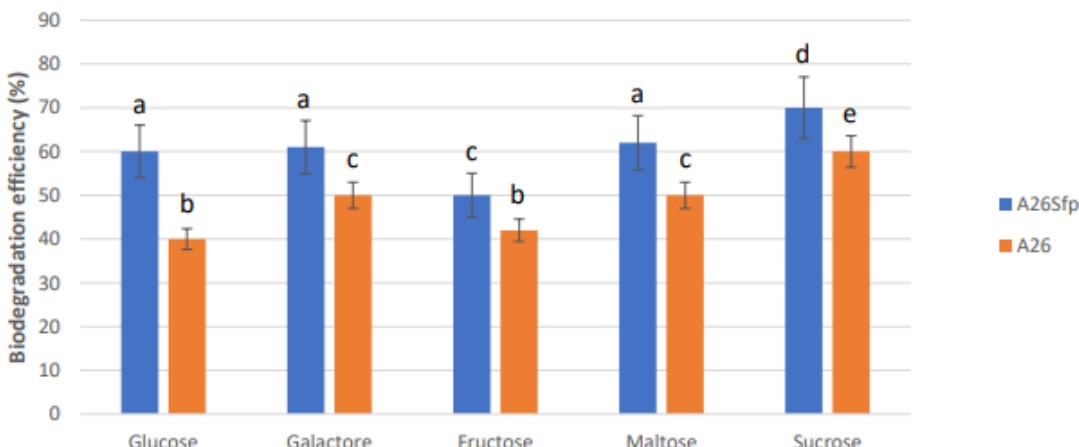

**Figure 3.** Effect of carbon sources on biodegradation of naphthalene *by Paenibacillus polymyxa* A26 and *P. polymyxa* A26Sfp. Different letters indicate statistically significant differences; see Material and Methods.

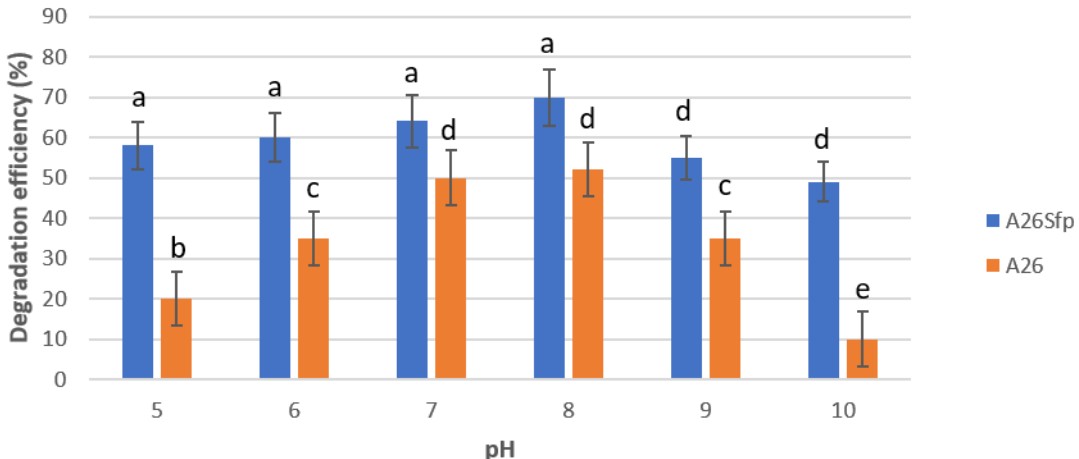

**Figure 4.** Effect of pH on biodegradation of naphthalene *by Paenibacillus polymyxa* A26 and *P. polymyxa* A26Sfp. Different letters indicate statistically significant differences; see Material and Methods.

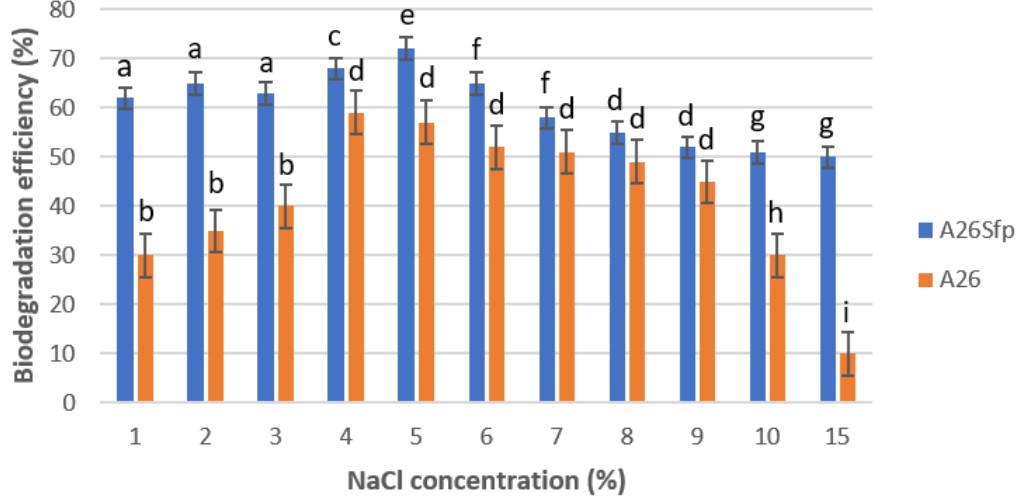

**Figure 5.** Effect of salinity on biodegradation of naphthalene *by Paenibacillus polymyxa* A26 and *P. polymyxa* A26Sfp. Different letters indicate statistically significant differences; see Material and Methods.

### 3.2. A26Sfp Ability to Emulsify Kerosene and Hexadecane Is Improved Compared to the Wild Type

The standard assay for emulsifying activity was based on a modification of the method of Cooper and Goldenberg [32,35]. Emulsifying activity was tested in both bacterial strain culture filtrates and the mutant glucan and EPS extracts on the 1st, 3rd, and 7th day. The tested compounds included kerosene and hexadecane, and the results are presented in Table 2. The results with A26 and A26Sfp culture filtrates, A26Sfp EPS and A26Sfp glucan extracts show 63%, 45%, 44%, 20%, degradation ability with hexadecane and 59%, 45%, 45%, 21% degradation ability with kerosene, respectively. (Table 2). While the mutant A26Sfp culture filtrates emulsified kerosene and hexadecane at 63% and 59%, respectively, the wild type A26 emulsification ability of kerosene and hexadecane did not differ significantly and was about 45%. The A26Sfp EPS extracts' emulsification ability activity was similar to that of the wild-type culture filtrates, while that of the A26Sfp glucan extracts was about 20% (Table 2). All treatments that showed emulsification activity were also positive in the drop and oil spreading assays (Table 2).

**Table 2.** Emulsion indices (%) of the strains using kerosene and hexadecane, glucan titre and other characteristics.

| Strains | Emulsion Index % Hexadecane (day) | | | Emulsion Index % Kerosene (day) | | | DC [1] | OS [2] | Glucan Titre (mg/L) | EPSTitre (mg/L) |
|---|---|---|---|---|---|---|---|---|---|---|
| | 1 | 3 | 7 | 1 | 3 | 7 | | | | |
| A26Sfp | 60 ± 4.2 [a] | 63 ± 5.2 [a] | 60 ± 5.2 [a] | 63 ± 5.2 [a] | 59 ± 5.2 [a] | 60 ± 4.1 [a] | + | + | 1.2 ± 0.1 | 15 ± 0.2 |
| A26 | 42 ± 5.2 [b] | 44 ± 5.2 [b] | 45 ± 3.2 [b] | 43 ± 5.2 [b] | 45 ± 2.2 [b] | 43 ± 3.2 [b] | + | + | 0.62 ± 0.1 | 10 ± 0.2 |
| EPS A26Sfp | 41 ± 2.2 [b] | 45 ± 2.2 [b] | 44 ± 2.2 [b] | 43 ± 2.2 [b] | 44 ± 3.2 [b] | 45 ± 2.2 [b] | + | + | ND | ND |
| Glucan A26Sfp | 20 ± 2.0 [c] | 20 ± 2.2 [c] | 21 ± 2.2 [c] | 21 ± 2.2 [c] | 21 ± 2.1 [c] | 23 ± 2.2 [c] | + | + | ND | ND |

[1] Drop-collapse test. [2] Oil spreading assay. [a–c] Different letters indicate statistically significant differences ($p < 0.05$).

### 3.3. Exopolysaccharide Glucan Production of A26Sfp Is Improved Compared to That of the Wild Type

The polysaccharide glucan production was estimated after weighing the lyophilised material produced by the A26 and A26Sfp strains. While the A26 grown in the minimal media produced glucans at 0.62 mg/L, the mutant ability was two times higher at 1.2 mg per L (Table 2).

## 4. Discussion

Our study reveals that A26Sfp and A26 efficiently degraded NAP by 69% and 49%, respectively, after 5 days of incubation in liquid medium and soil (Figures 1 and 2a,b). Somewhat reduced numbers for the degradation efficiency in the case of degradation in soil reflect that additional factors such as soil physical properties and texture porosity may influence the degradation process (Figure 2a,b) [35]. Generally, microorganisms require suitable growth conditions (e.g., as regards carbon sources, nutrients, pH) which strongly affect their growth. Carbon sources in the growth medium are considered to be key factors for PAH-degrading bacteria. For use in bioremediation, PAH degraders should ideally mineralize and grow on PAHs as sole carbon and energy source. This would minimize the production of toxic, water-soluble degradation by-products. In our experiments, the strains that used sucrose and 5% NaCl were most efficient at pH 7 and 8 (Figures 4 and 5). pH is one of the most important factors for PAH degradation in the culture medium, as it affects bacterial enzymatic activity as well as nutrient solubility [8,36,37]. It is interesting that the mutant strain can remain effective even at extreme conditions of pH and salinity (Figures 4 and 5). While bacterial growth and degradation usually prefer neutral pH, the waste environments often are alkaline and some degradation enzymes may be only activated under alkaline conditions [38,39].

Most often, chemical parameters are employed to evaluate and analyse PAHs in the environment. The chemical methods are accurate and sensitive for specific molecules but do not give information regarding biological influence within the ecosystem. Chemical

methods do not consider synergistic effects from compound mixtures and there is therefore a risk that they underestimate the toxic potential of a particular sample. Therefore, evaluation of biological influence using rapid, simple, and economic methods can provide information about all compounds and incorporate these important toxicity parameters. The bioluminescence inhibition method has been shown to correlate well with the total level of PAHs and has been used for NAP ecotoxicity studies [40].

Several microbes have a great potential for the production of bioactive secondary metabolites associated with PAH biodegradation, and various pathways have been revealed [41–44]. However, owing to the hydrophobic nature of the PAHs, the first step must be surface tension reduction [6,37]. Therefore, the biodegradation of PAH is dependent on biosurfactants, i.e., compounds that lower interfacial tension between PAH and the soil water face. Different classes of surfactants have been discovered, and the surfactin group encoded by the *sfp* gene belongs to the group of non-ribosomal compounds that has been considered as one of the most powerful biosurfactant groups [45]. In our study, the bacterial surfactin production was inactivated, as non-ribosomal proteins including surfactin are not produced by the mutant strain A26Sfp [12]. In contrast to surfactin production, the EPS and glucan were surprisingly overproduced by the mutant strain, and A26Sfp emulsification ability can be linked with, but not limited to, its EPS and glucan production (Table 2). The compounds showed significant ability to emulsify the NAP and reduce tension between NAP and the soil water (Table 2). The A26Sfp emulsification ability is comparable to a 0.5 solution of the non-ionic surfactant Tween 80 [36]. Polysaccharide surfactants are an emerging class of biodegradable nontoxic and sustainable alternatives to conventional surfactant systems [3,4,46,47].

Here, we show that, besides non-ribosomally produced lipopeptides of P. polymyxa (represented by the sfp gene products), there are other most-efficient EPS surfactants produced by A26Sfp. Polysaccharide-based emulsifiers of microbial origin have attracted attention, as they offer several advantages over synthetic emulsifiers, including lower toxicity, higher degradability, and better compatibility with the environment. In addition, they can remain effective even at extreme conditions of pH and salinity. These properties increase their scope for application in a diverse range of biotechnological areas. It has been repeatedly shown that the bacteria from extreme environments offer good candidates for efficient PAH biodegradation and biosurfactant producers [9]. To the best of our knowledge, it has not been reported earlier that Sfp-type 4-phosphopantetheinyl transferase deletion, making the strain incapable of NRP and PKS production, can cause a significant change in polysaccharide production and composition.

## 5. Conclusions

Here, we show that the *P. polymyxa* A26 Sfp-type 4-phosphopantetheinyl transferase deletion causes a significant change in polysaccharide production and composition. The polysaccharide-based surfactants of the mutant carry significant abilities to emulsify the polycyclic aromatic hydrocarbons. While the wild type A26 biodegradation efficiency is significantly reduced at extreme conditions of pH and salinity, the mutant strain remains effective and is of importance for the soil and water remediation industry. A26Sfp is easy and safe to use, as the mutant is non-sporulating, which abolishes the possibility for secondary contamination. The rich composition of the *P. polymyxa* A26Sfp exopolysaccharides and activity at extreme conditions (pH and salinity) deserves further attention.

**Supplementary Materials:** The following are available online at https://www.mdpi.com/article/10.3390/stresses1040019/s1, Figure S1. The Evolution Canyon (EC) model. Schematic diagram of the Evolution Canyon at Lower Nahal Oren, Mount Carmel (source 60 Nevo, 2012 Evolution Canyon, a potential microscale monitor of global warming across life, PNAS 109; 8) (Photo by S. Timmusk).

**Author Contributions:** Conceptualization, S.T.; methodology, S.T. and T.T.; validation, S.T., T.T. and L.B.; writing—original draft preparation, S.T. writing—review and editing, S.T., T.T. and L.B.; visualization, L.B. All authors have read and agreed to the published version of the manuscript.

**Funding:** This research was funded by the Swedish Research Council 2017-522.

**Institutional Review Board Statement:** Not applicable.

**Informed Consent Statement:** Not applicable.

**Data Availability Statement:** The data presented in this study are available in article or supplementary material.

**Acknowledgments:** We are grateful for David Clapham for critically reading the manuscript.

**Conflicts of Interest:** The authors declare no conflict of interest.

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
