# Peer review of "Paenibacillus polymyxa A26 and Its Surfactant-Deficient Mutant Degradation of Polycyclic Aromatic Hydrocarbons"

_stresses, doi:10.3390/stresses1040019_

Round 1

Reviewer 1 Report

Line 24, page 1: Name the major and minor constituents

Line 47, page 2: Repetition of the words must be deleted, “is isolated from the 47 stressful SFS”

Line 88, page 2: sub-heading must be comprehensive and needs to be rewritten“Analysis of NAP biodegradation efficiency by Aliivibrio fischeri luminescence inhibition biotest”

Line 122, sub-heading 2.4 contains the same sentences as mentioned in the sub-heading 2.3, can’t we use a single subheading which covers both the tests mentioned in 2.3 and 2.4?

Line 129, page 3: The sentence “shaking incubator as described above for” is repeated in every section of methodology. Can’t we tell the rpm of the incubator?

Line 132, page 3: Please elaborate in which section of the manuscript, this part is mentioned, “as described earlier with small modifications”

Line 142, page 4: Subsection 2.8, Please tell the section where it is mentioned, “The bacterial strains were incubated for 5 days as described above”

Line 179, page 4: Recheck the number of cells, “106 cells”

Line 181, page 4: Where to see, mention the section in sentence, “pH 8 (see above)”

Line 184, page 4: Mention the time period in sentence, “periodically”

Line 188, page 4: The reaction was performed in 10 ml of what?

Line 197, page 5: check spacing between fill stop and the words.

Figure 2a: What is nedium mentioned in the sentence given nelow the figure 2a

Line 216. Check the punctuation. Moreover, is there any need to mention the sentence about material and methods? If so, then mention the section or sub-heading to check

Line 236-237: The sentence is incomplete, “Emulsifying activity was tested in both bacterial strain culture filtrates as well as the 236 A26Sfp EPS and A26Sfp glucan extracts”

Line 255: Put some citations if available for the sentence, “texture porosity may influence the degradation process”

263-264: Can you explain how nutrients are available even at pH 8 and which enzymes work are present at this pH level and work for degradation and nutrient availability as well

Line 281-299: The font size must be symmetrical to the whole manuscript. Moreover, punctuation errors must be corrected like many places, double full stop is pressed which needs attention.

Conclusion needs attention. Please mention which industry can be benefitted and what are the distinguished results which were obtained in the research as well as what is its novelty

Author Response

The authors would like to thank the reviewer for your kind and comprehensive comments, which facilitated our efforts in making the proper improvements to the manuscript. Unfortunately, the line numbers that were indicated for the revisions, did not match up with manuscript. We are hoping that the modifications made are in the proper places. Please accept our sincerest apologies for the lateness. Our workload has been overwhelming, which was intensified with the usual seasonal ‘colds’ and time away from work. Your understanding is much appreciated, and we are doubling our efforts to catch up.

Line 24, page 1: Name the major and minor constituents

Corrected in the manuscript

Line 47, page 2: Repetition of the words must be deleted, “is isolated from the 47 stressful SFS”

Corrected in the manuscript

Line 88, page 2: sub-heading must be comprehensive and needs to be rewritten“Analysis of NAP biodegradation efficiency by Aliivibrio fischeri luminescence inhibition biotest”

The subtitle is replaced with the new subtitle

Line 122, sub-heading 2.4 contains the same sentences as mentioned in the sub-heading 2.3, can’t we use a single subheading which covers both the tests mentioned in 2.3 and 2.4?

The section 2 subheadings are modified

Line 129, page 3: The sentence “shaking incubator as described above for” is repeated in every section of methodology. Can’t we tell the rpm of the incubator?

The Material and Methods section is rewritten for clarity

Line 132, page 3: Please elaborate in which section of the manuscript, this part is mentioned, “as described earlier with small modifications”. Unfortunately, we didn’t understand the comment

Line 142, page 4: Subsection 2.8, Please tell the section where it is mentioned, “The bacterial strains were incubated for 5 days as described above”

The Material and Methods section is rewritten

Line 179, page 4: Recheck the number of cells, “106 cells”

Corrected in the manuscript

Line 181, page 4: Where to see, mention the section in sentence, “pH 8 (see above)”

The Material and Methods section is rewritten

Line 184, page 4: Mention the time period in sentence, “periodically”

Corrected in the manuscript

Line 188, page 4: The reaction was performed in 10 ml of what?

Corrected in the manuscript

Line 197, page 5: check spacing between fill stop and the words.

Corrected in the manuscript

Figure 2a: What is nedium mentioned in the sentence given nelow the figure 2a

Corrected in the manuscript

Line 216. Check the punctuation. Moreover, is there any need to mention the sentence about material and methods? If so, then mention the section or sub-heading to check. Sorry, could you please rephase the comment.

Line 236-237: The sentence is incomplete, “Emulsifying activity was tested in both bacterial strain culture filtrates as well as the 236 A26Sfp EPS and A26Sfp glucan extracts”

Corrected in the manuscript

Line 255: Put some citations if available for the sentence, “texture porosity may influence the degradation process”

Corrected in the manuscript

263-264: Can you explain how nutrients are available even at pH 8 and which enzymes work are present at this pH level and work for degradation and nutrient availability as well

Unfortunately, an error has happened in the manuscript. After rechecking the Figure, we discovered that while there is a slight increase in degradation efficiency at pH 8, it is not statistically significant, see line 224-227.Our results show that the two strains used in the study perform best in the medium supplemented with sucrose (Fig 3) and 5% NaCl (Fig 5) at pH 7 and 8 (Fig 4). While the A26 biodegradation efficiency is significantly reduced at the pH 5, 6, and at 15% NaCl, the mutant   remains effective even at extreme conditions of pH and salinity (Fig 4 and 5).  

 The focus of the work however in not on the peak at pH 7/8 . Rather it is on the wide range of activity at extreme conditions. Unfortunately, we don't have any details, since there are a number reasons that could be involved. This is an interesting issue that deserves further study.  

Line 312-314 The rich composition of the P. polymyxa A26Sfp exopolysaccharides and activity at extreme conditions (pH and salinity) deserves further attention.  

Line 281-299: The font size must be symmetrical to the whole manuscript. Moreover, punctuation errors must be corrected like many places, double full stop is pressed which needs attention.

Corrected in the manuscript

Conclusion needs attention. Please mention which industry can be benefitted and what are the distinguished results which were obtained in the research as well as what is its novelty

Conclusion is rewritten

Reviewer 2 Report

The manuscript can be of interest but a major revision is required, especially for a clearer and detailed description of the presented results.

Figure 1, 2 , 3, 4 and 5 are not sufficiently described in text of the manuscript.

Results presented in Table 2 are not sufficiently described in the text. Moreover, please define OC and OS in table 2.

Please revise 3.1, 3.2 and 3.3 paragraphs (results section).

Author Response

The authors would like to thank the reviewer for your kind comments, which facilitated our efforts in making the proper improvements to the manuscript. Please accept our sincerest apologies for the lateness. Our workload has been overwhelming, which was intensified with the usual seasonal ‘colds’ and time away from work. Your understanding is much appreciated, and we are doubling our efforts to catch up.

Figure 1, 2, 3, 4 and 5 are not sufficiently described in text of the manuscript.

The Results section is rewritten. Several sections in the Discussion are rewritten.

Results presented in Table 2 are not sufficiently described in the text. Moreover, please define OC and OS in table 2.

Explanations on Table 2 have been introduced to the Discussion and Results section.

Please revise 3.1, 3.2 and 3.3 paragraphs (results section).

Section 3 is rewritten in the manuscript.

Round 2

Reviewer 2 Report

The manuscript is suitable for publication